# Modulation of Notch Signaling by Small-Molecular Compounds and Its Potential in Anticancer Studies

**DOI:** 10.3390/cancers15184563

**Published:** 2023-09-14

**Authors:** Arkadiusz Czerwonka, Joanna Kałafut, Matthias Nees

**Affiliations:** Department of Biochemistry and Molecular Biology, Medical University of Lublin, 20-093 Lublin, Poland; joanna.kalafut@umlub.pl (J.K.); matthias.nees@umlub.pl (M.N.)

**Keywords:** notch signaling, cancer, small-molecular compounds

## Abstract

**Simple Summary:**

The dysregulation of Notch signaling plays an important role in cancer development. Numerous attempts are made to modulate Notch signaling for positive therapeutic effects. For this purpose, a broad group of small-molecular compounds are used. These small compounds have several features desirable in laboratory and clinical practice, such as low price, easy availability, and the possibility of standardizing the dose, and will be presented in this article.

**Abstract:**

Notch signaling is responsible for conveying messages between cells through direct contact, playing a pivotal role in tissue development and homeostasis. The modulation of Notch-related processes, such as cell growth, differentiation, viability, and cell fate, offer opportunities to better understand and prevent disease progression, including cancer. Currently, research efforts are mainly focused on attempts to inhibit Notch signaling in tumors with strong oncogenic, gain-of-function (GoF) or hyperactivation of Notch signaling. The goal is to reduce the growth and proliferation of cancer cells, interfere with neo-angiogenesis, increase chemosensitivity, potentially target cancer stem cells, tumor dormancy, and invasion, and induce apoptosis. Attempts to pharmacologically enhance or restore disturbed Notch signaling for anticancer therapies are less frequent. However, in some cancer types, such as squamous cell carcinomas, preferentially, loss-of-function (LoF) mutations have been confirmed, and restoring but not blocking Notch functions may be beneficial for therapy. The modulation of Notch signaling can be performed at several key levels related to NOTCH receptor expression, translation, posttranslational (proteolytic) processing, glycosylation, transport, and activation. This further includes blocking the interaction with Notch-related nuclear DNA transcription. Examples of small-molecular chemical compounds, that modulate individual elements of Notch signaling at the mentioned levels, have been described in the recent literature.

## 1. Canonical and Non-Canonical Notch Pathway

Notch signaling (Figure 1) in mammals is based on the activation of one of the four single membrane pass NOTCH mechanoreceptors (NOTCH1, 2, 3, and 4) by their transmembrane ligands, Delta-like-1,3,4 (DLL1; 3; and 4) or Jagged 1, 2 (JAG1 and JAG2), collectively known as Delta/Serrate/Lag-2 (DSL) ligands. The basic structure of the NOTCH receptor can be distinguished as (a) the NOTCH extracellular domain (NECD) with the region containing a variable number of EGF-like repeats responsible for receptor–ligand interaction and (b) the negative regulatory region (NRR) with the S1 and S2 cleavage sites; followed by (c) a short transmembrane domain (TM) which contains the critical S3 cleavage side, and (d), the highly relevant NOTCH intracellular domain (NICD). This basic architecture is similar in all four NOTCH receptors, although the length of these domains and, for example, the number of EGF-like domains, varies. The functionally essential NICD of all four NOTCH receptors is itself composed of several distinguished domains, including an RBPjκ association module (RAM), ankyrin repeats (ANK; flanked by nuclear localization signal (NLS) sequences), the transactivation domain (TAD), and proline/glutamic acid/serine/threonine-rich motifs (PEST), responsible for interaction with transcription factors (RAM, ANK), regulation (TAD), and protein degradation (PEST). NOTCH ligands usually have a much simpler structure, consisting only of a single membrane-anchored extracellular domain made of EGF-like repeats region and DSL domains [1,2,3,4].

After translation, the NOTCH proteins traffic to the plasma membrane via the endoplasmic reticulum (ER) and the Golgi apparatus [5], and undergo a series of post-translational modifications (PTMs), which are essential for the specificity of ligand recognition, and subsequently, the strength and functionality of NOTCH–DSL interactions. PTMs involve, among others, the addition of O-glycans (O-glycosylation and O-xylosylation by POGLUT1, O-fucosylation by POFUT1, O-GlcNAcylation by EOGT), all linked to the EGF-like extracellular domains of the receptors [6,7]. The key factors for Notch glycosylation are the Fringe family of proteins (lunatic fringe, manic fringe, and radical fringe) [8]. These are glycosyltransferases that transfer N-acetylglucosamine to the O-linked fucose of all four Notch receptors. Fringe proteins can act both as tumor suppressors and oncogenes, depending on cancer type, cancer progression stage, and probably unknown factors such as the expression of the different fringe proteins in individual cells and tissues, or the binding of different ligands and cell–cell interactions. This illustrates the complexity of Notch signaling and its regulation. In addition, even before the insertion into the plasma membrane, the first proteolytic cleavage (S1) of the maturing NOTCH receptors occurs, via Furin-like convertase. Also, this already occurs within the compartments of the Golgi apparatus [4].

The NOTCH receptor/ligand interaction, a mechanosensory event initiated by cell-cell contacts, triggers the endocytosis of ligands, which creates a pulling force and consequently initiates the canonical Notch pathway [9,10]. In the absence of a ligand, NOTCH receptors are unable to further activate, because the activating cleavage site (S2), is in an autoinhibited conformation state inside NRR [11]. After receptor–ligand binding, the DSL-NECD complex undergoes endocytosis in the signaling sending cell. This step exerts a pulling force which allows the unfolding and exposure of the S2 site to proteolytic cleavage by ADAM protease (S2 cleavage; mainly ADAM10 and 17), followed by a “final cut” mediated by the γ secretase complex (γS; S3 cleavage). This last proteolytic cleavage results in the release of the NICD into the cytoplasm, followed by its translocation to the nucleus [2]. Once in the nucleus, the NICD binds to the DNA-binding protein RBP-J, (recombination signal binding protein for immunoglobulin kappa J region; also known as CSL, CBF-1, Su(H) or Lag-1). In the absence of the NICD, RBP-J binds several transcriptional co-repressors and epigenetic players, such as histone deacetylases (HDACs), which prevents the active transcription of Notch target genes. The presence of the NICD protein enables the dissociation of these co-repressors from RBP-J, which allows the binding of an essential co-activator, Mastermind-like protein 1 (MAML1), and the formation of a ternary complex (known as the Notch ternary complex; NTC). The NTC acts as a scaffold for binding additional transcriptional co-activators and enzymes related to DNA binding and epigenetic loosening, such as the p300 family of histone acetyltransferases (HATs, including EP300).

The high flexibility of this system, related to the possibility of dynamic interactions of the NTC with a large number of additional regulatory proteins, enables the strict control of the transcription of “classic” NOTCH target genes, such as the Hes/Hey family of transcriptional repressors, CCND1 (Cyclin D1) and MYC, by binding to their promoter or enhancer region [10,12,13,14]. It is also important to consider that the four NOTCH receptors have partially overlapping but also unique functions, and that they activate a different spectrum of target genes. The four NOTCH receptors further show markedly different cell- and tissue-type specific expression, and the spectrum of Notch target genes strongly depends on the cell and tissue type under investigation [15]. The transcriptional targets of Notch signaling are largely context-specific, and many of these genes are even located in tissue-specific enhancers [16], which allows a differential function that depends on context and even time. Thus, Notch signaling depends to a large degree on context [17] and (micro)environment, cell–cell contacts, and cell density in various tissues. As a mechanoreceptor, it is also likely that even aspects such as the composition, density, stiffness, and pressure of the extracellular matrix strongly affect Notch pathway activity, quality, strength, and the direction of signaling.

In addition, non-canonical pathways [18,19] of Notch signaling that bypass RBP-Jκ and MAML have been reported mainly in immune cells, and less frequently in solid cancers, although their mechanisms and relative importance remain debated [20,21]. Also, the non-canonical Notch pathway is associated with NICD activity, although it does not use the formation of the NTC complex. In variations of this non-canonical Notch pathway, the interaction of the NICD may occur with various signal molecules, such as AKT, mTOR, NF-κB [22], or HIF-1α [23], ultimately leading to changes in the expression of the usual broad spectrum of target genes. Interestingly, the activation of the non-canonical pathway may also be triggered by a non-canonical ligand, or it can occur in the absence of a ligand, and it may not even require NOTCH receptor cleavage [9,24]. The role of the non-canonical Notch pathway and its impact on the development and progression of cancer is still an intensively studied area that may shed new light on the possibilities of functional Notch modulation in the treatment of cancer [25] in the future. Understanding Notch signaling allows researchers to interfere with its key structures and functional connections, which may achieve specific biological effects and processes that are involved in the progression of diseases such as cancer.

Generally, Notch can play an essential, but pleiotropic, role during various stages of carcinogenesis and cancer progression, with highly tissue-specific and context-dependent outcomes, especially when considering the cross-talk with other signaling pathways, such as Wnt and Hedgehog [26]. Nevertheless, the dysregulation of Notch signaling is observed in a wide spectrum of cancers, which proved a significant incentive for interfering with it as a therapeutic target.

## 2. Two Sides of the Same Coin—Notch Gain- vs. Loss-of-Function

Notch signaling is an evolutionarily conserved pathway that relies on signal transduction between neighboring cells. The Notch system enables the control of the spatial localization of cells in context, and their development into organized tissue structures. During embryogenesis, Notch (together with other pathways such as Wnt and Hedgehog) is responsible for the control of somitogenesis and, furthermore, for the development of the dermis, muscles, vascular, skeleton, and nervous systems [27,28]. Notch signaling also plays a key role in maintaining tissue homeostasis after birth and maturity, by controlling processes such as tissue turnover and regeneration, wound healing, and immune response [27,29,30].

The dysregulation of Notch signaling plays an important role in cancer development. Notch signaling controls basic processes related to cancer initiation and development, such as chemo- [31,32], immune- [33,34], and radio-resistance [35], tumor cell plasticity, epithelial-to-mesenchymal transition (EMT) [36,37,38], angiogenesis [39], and the maintenance of a cancer stem-cell-like phenotype [40,41,42]. Hyperactivated Notch signaling may, thus, also play a role in tumor dormancy, relapse, and local and distant metastasis. In general, two fundamentally opposing patterns of dysregulation of Notch signaling are observed in cancers: oncogenic hyper-activation (or gain-of-function, GoF mutations), versus tumor suppressor function (due to loss-of-function, or LoF mutations). Activating and oncogenic GoF mutations in the NOTCH receptors (mostly NOTCH1) typically play a role in promoting later-stage tumor progression but can also act as gatekeepers or lineage-specific driver mutations involved in cancer initiation, such as in T-ALL [43]. Genome-wide mutation analyses from the past two decades suggest a strong, causal connection between a hyperactivated Notch pathway and the development of certain types of cancer, including acute myeloid leukemia (AML) [44], bladder cancer (BC) [45], adenoid cystic carcinoma (ACC) [46], and, above all, T-cell acute lymphoblastic leukemia (T-ALL) [43,47,48,49]. Activating NOTCH receptor mutations are also frequently found in advanced, triple-negative breast cancer (TNBC) [50,51,52,53]. Alterations in the amplification and overexpression of Notch receptors are prevalent in breast cancer. Predominantly, these alterations involve NOTCH2 (up to 13%) [54] and NOTCH1 (2–5%), while occurrences are slightly less frequent for NOTCH3 and NOTCH4 (2–3%) [55,56]. The hyperactivation and/or overexpression of NOTCH receptors typically results in the strong, ligand-independent activation of the receptor, while GoF mutations typically increase the lifespan of the functional domain of the receptor (Notch intracellular domain; NICD [3]). In addition, mutations in other genes associated with the Notch pathway, like AJUBA [57] and FBXW7 [58], can lead to the altered lifespan of NOTCH proteins and modulation of the functions of the NICD. AJUBA, for example, modulates Notch signaling by the sequestration of the NUMB protein, which promotes effective nuclear translocation of the NICD to regulate gene expression. It is related to the fact that NUMB is able to directly inhibit Notch functions by promoting the degradation and polyubiquitination of Notch proteins [59]. Surprisingly, it has recently been discovered that NUMB can enhance Notch signaling by regulating the proteasome pathway. This is related to the fact that NUMB, by controlling the activity of BAP1 deubiquitinase (BRCA1-related protein 1) and BRCA1 (Breast Cancer type 1 susceptibility protein), facilitates the stabilization of NICD [60].

FBXW7 is a F-box-containing protein that acts mainly as a tumor suppressor. FBXW7 regulates the activity of a range of substrates, including NOTCH1, through degradation by the ubiquitin-proteasome system. Functional, active FBXW7 regulates NOTCH protein activity by controlling its half-life, maintaining optimum protein levels in cells. LoF mutations of FBXW7, which are common in head and neck cancers, thus lead to an extended half-life and hyperactivity of the NICD [61]. Despite these important roles in Notch signaling, no small-molecule inhibitors have yet been developed for targeting FBXW7 or AJUBA.

LoF mutations in NOTCH receptors are characteristic of squamous cell carcinomas (SCC) such as head and neck squamous cell carcinoma (HNSCC) [62], lung squamous cell carcinoma (LSCC), and cutaneous squamous cell carcinoma (CSCC) [63,64]. They may act as gatekeeping mutations early in tumor initiation and the first stages of progression [65]. In these cases, NOTCH receptors primarily act as tumor suppressor [66] genes, as reviewed in references [17,67]. It is assumed that NOTCH receptors essentially act as tumor suppressors in squamous epithelial cells and may be required for the terminal (squamous) differentiation of the skin, cervical or oral mucosa [68]. The loss of Notch signaling likely promotes the further development of tumors in the early stages of dysplastic tissues (e.g., in virus-infected cells or cells exposed to UV light or chemical carcinogens) by impairing the balance between proliferation, terminal squamous cell differentiation, and pro-apoptotic signaling. This may, consequently, favor the maintenance and proliferation of mutated stem/progenitor cells which, along with other mutations, could then further contribute to initiating the process of carcinogenesis [69,70]. This hypothesis is supported by the observation that γ-secretase inhibitors, a class of drugs that reduce Notch signaling, may increase the risk of skin cancer development in patients who have received any of these drugs in clinical studies [71]. In HNSCC, truncating, missense, and splice site mutations in NOTCH1 that result in non-functional proteins are found in up to 20% of the cases [54]. The NOTCH1 mutation frequency is significantly higher in cancers that are negative for human papillomaviruses (HPV), indicating a different etiology of these distinct cancer subtypes [72,73]. LoF mutations in other NOTCH receptors are somewhat less frequently found, e.g., for NOTCH2 in 5–8% of HNSCC, for NOTCH3 in 3–5% of the cases, and only 2–3% in NOTCH4. Taken together, around 35–40% of all cases show aberrations in NOTCH receptors [74,75,76] (according to cBioPortal for Cancer Genomics (https://www.cbioportal.org (accessed on 10 May 2023) and NCI Genomic Data Commons (https://gdc.cancer.gov (accessed on 10 May 2023). The loss of Notch signaling in HNSCC is also reviewed in several publications [66,68,77], which also mention the potential bimodal characteristics of Notch mutations at later stages of cancer progression. The presence and role of gain-of-function alterations, such as overexpression in NOTCH receptors in HNSCC and other squamous carcinomas, remains debated. GoF mutations, such as those typical in T-ALL, are not found in HNSCC, and functional alterations are mainly related to high expression levels. Recent publications have reported a diagnostic and prognostic association of high-level NOTCH receptor expression in HNSCC [78] in European patients. Other reports of GoF alteration in Notch signaling components, including the ligands, have been predominantly observed in Asian patient cohorts, in which many suffer from different risk factors compared to European and American patients [79]. There are also reports of GoF mutations in HNSCC or other squamous carcinomas [79] in advanced, recurrent, or metastatic cancers. This suggests a potential biphasic or bimodal role of Notch pathway activity and mutations in cancer progression [77,80], in which the initially silenced pathway may be re-activated and even hyperactivated at later progression stages [80,81,82]. The relevance of this for cancer therapy and patient outcome is still poorly understood. Future research may need to also consider the aberrant expression of Notch receptors and ligands in the tumor microenvironment, such as in cancer-associated fibroblasts (CAFs), lymph vessels, and blood vessel endothelial cells. Generally, Notch signaling in different cell types within the tumor tissues is likely to significantly contribute to the intra-tumor heterogeneity, but is poorly investigated.

If additional, Notch-pathway-related genes such as AJUBA, EP300, and FBXW7 are included (each mutated with a frequency of 7–8% in HNSCC) and it appears that about two-thirds or >60% of all HNSCC harbor changes in Notch signaling [83]. Furthermore, a substantial number of tumors exhibit epigenetic silencing of the second allele of NOTCH1 located on chromosome 1 [52]. Consequently, this leads to a significant reduction in both the expression and functionality of this receptor. Epigenetic silencing is not routinely observed for the other NOTCH receptors. The degree to which the four NOTCH receptors may be able to functionally compensate for each other remains unclear, but is to be considered a possibility that requires further research. In the past, this has been subject mainly to studies in developmental biology [84], but not in oncology. In particular, NOTCH1 and 2 have been demonstrated to, at least partially, compensate for each other [85], for example, in maintaining the functional stem cell niche in gastrointestinal crypt structures [86]. There are no comparable studies in cancers.

Mutations in Notch ligands are found more rarely in the same cancers, ranging only between 0.5 and 3% for DLL1, DLL3, DLL4, JAG1, or JAG2. Most of these genetic aberrations are amplifications that can lead to the overexpression of the ligands; while inactivating or truncating mutations are very rare (<0.5%; according to The Cancer Genome Atlas Program (TCGA)—NCI; https://www.cancer.gov (accessed on 26 May 2023). In addition, about 30% of HNSCC or skin cancer cases show overexpression of JAG1 or JAG2 [87], which may be in line with the activation of the Notch pathway at least in some cases [88,89], especially aggressive, late-stage cancers, and is associated with poor patient outcome [88,89].

Finally, there is a broad group of cancers such as the brain [90], urothelial [3], or small-cell lung cancers (SCLC) [91,92] in which alterations of the Notch pathway may be functionally involved, but their precise role is ambiguous and still far from being understood. In SCLC, it is even possible that both gain- and loss-of-function mutations occur simultaneously, side by side, within different cells of the same tumor tissue. Both neuroendocrine cancer cells, and cells with more epithelial differentiation, which coexist in this cancer side by side, showed inactivating and activating NOTCH1 mutations, respectively. This resulted in the mutual support and cooperation between these cell types in tumor progression and led to markedly differential sensitivity to chemotherapy. This demonstrates how critically Notch signaling and pathway activity depends on context [10], environment, extracellular matrix, and cell–cell contacts.

Furthermore, targeting the Notch signaling pathway may likely only benefit a fraction of cancer patients with a certain tumor type. This almost entirely depends on the presence or absence and functional impairment of Notch-related mutations, the genetic background of the patients, but also context- and tissue-specific parameters within individual tumors, such as activity of stromal components. Thus, targeting Notch signaling using chemotherapy in patients represents an issue with an outstanding individualized or personalized character. Interfering with Notch signaling on behalf of cancer chemotherapy may be a prototype case for future personalized cancer medicine; each patient will likely show a unique response pattern to Notch inhibitors, especially if combined with additional targeted drugs such as PI3Kinase, AKT, mTOR, and other inhibitors [25,93]. Whether any treatment with Notch-related drugs will be beneficial for patients remains to be tested ex vivo/in vitro, for example, based on physiologically relevant tumor models, such as patient-derived organoids, explants, or xenografts. In the following paragraphs, we address the most frequently used mechanisms of action exploited by small-molecule inhibitors to interfere with Notch signaling activity.

## 3. Compounds Interfering with Endoplasmic Reticulum-to-Golgi and Membrane Trafficking of NOTCH Receptors

Disturbing the trafficking of premature and proteolytically processed forms of the NOTCH receptors prevents the stepwise maturation and transport of NOTCH receptors to the plasma membrane, impairing Notch signaling (Figure 2). Several small molecular weight compounds have shown significant activity in this area, particularly in inhibiting Notch signaling in cases of T-ALL models. NVS-ZP7-4 is an inhibitor of the zinc-ion transporter SLC39A7 (ZIP7) that blocks the release of Zn^2+^ into the cytosol from ER/Golgi structures. The treatment of T-ALL cells with NVS-ZP7-4 blocked the intracellular transport of premature NOTCH proteins via ER stress, ultimately leading to apoptosis [94]. A similar model of action related to interfering with calcium efflux has recently been described. Inhibition of the calcium sarco/endoplasmic reticulum calcium ATPase (SERCA) transporter by small-molecular weight molecules, such as Thapsigargin [95,96], and Casearin J [97] impaired the correct folding and maturation of premature NOTCH receptors, and induced G_0_/G_1_ arrest in T-ALL cells. SERCA was first identified by genomic screens as a potential target in cancer cells that harbor mutated NOTCH1 [95]. This discovery has prompted the development of novel pharmacological inhibitors targeting SERCA. These inhibitors exhibit improved drug-like characteristics, fewer off-target effects, and reduced cytotoxicity in comparison to Thapsigargin. The compound CAD204520 is a small-molecule inhibitor of SERCA with reduced off-target Ca^2+^ toxicity that is currently in clinical trials against T-ALL [98]. The most recent developments of SERCA inhibitors targeting Notch signaling are summarized by Pagliaro et al., 2021 [99] and 2020 [100]. Other calcium transporter blockers, such as the non-selective drug Bepridil, are also suspected to show similar features [101] as described for Thapsigargin.

The inhibition of NOTCH secretion before its exit from the ER and entry into the Golgi apparatus was observed with dihydropyridine derivatives such as FLI-06 [102]. FLI-06 effectively inhibits Notch signaling in lung adenocarcinoma [103] and HNSCC [104,105]. FLI-06 treatment is associated with G_1_ cell cycle arrest and programmed cell death/apoptosis. The inhibition of NOTCH receptor trafficking has also been reported for Chloroquine, originally developed as one of the first antimalarial drugs, showing the ability to inhibit lysosome function and autophagy. Along these lines, the chloroquine-induced accumulation of the NOTCH1 receptor in the autophagosome was observed [106]. Similarly, Bafilomycin A1 (BafA1), a highly specific Vacuolar H^+^ ATPase (V-ATPase) inhibitor, leads to the accumulation of NOTCH in the endo-lysosomal system, and consequently, reduces Notch signaling in normal breast cells. Interestingly, in a breast cancer model, BafA1 treatment reduces the growth in cells expressing membrane-tethered, constitutively active NOTCH1 forms, while sparing cells expressing cytoplasmic forms [107].

Blocking membrane trafficking strongly inhibits Notch signaling due to the reduced exposure and density of NOTCH receptors on the cell surface. However, these effects are mostly related to the indirect reduction in all four types of NOTCH receptors. Not only does pan-NOTCH inhibition (blocking all four types of NOTCH receptors) complicate the research on the role of individual types of receptors, it also makes it more difficult to achieve the desired therapeutic effects. Additionally, membrane trafficking inhibition, the blocking of ion channels, and interfering with intracellular ion concentrations are all notoriously non-specific mechanisms of action. This line of research carries a high risk of non-specific interference with the transport of many other membrane proteins, and ion-dependent signaling pathways, thus making the interpretation of the results related to Notch signaling difficult.

## 4. NOTCH–Ligand Interaction, Γ-Secretase, and ADAM Proteases Inhibition

Notch signaling can also be modulated by modifying the sugar residues of EGF-like extracellular domains of the NECD. The inhibition of Notch signaling can be accomplished with O-fucose analogs such as 6-alkynyl and 6-alkenyl fucose, specifically by incorporating these analogs into EGF-like domain 8 (EGF8), which disrupts the DLL-NECD interaction. Interestingly, both of these analogs do not disturb Jagged-induced, Notch signaling [108]. This observation opens up a new field of study for different O-glycans to fine-tune the strength of Notch signaling by the modification of the sugar residues of the NICD.

The most widely used mechanism of action that is employed by small-molecule inhibitors relates to the proteolytic processing of NOTCH receptors (Figure 3), especially by the gamma secretase complex. γ-Secretase (γS) is a multifactor complex formed by presenilin 1/2 (homolog PS1 or PS2), nicastrin (NCT), anterior pharynx-defective 1A/1B (APH-1a or APH-1b), and presenilin enhancer 2 (PEN-2) proteins. Anchored in the cell membrane, γS is responsible for the hydrolysis of substrates through the activity of its catalytic subunit, presenilin. The proteolytic cut of the S3 cleavage site of all four NOTCH receptors represents a key event in the activation of canonical Notch signaling [109,110]. Thus, the inhibition of the γ-secretase complex by γ-secretase inhibitors (γSI or GSI) is the most intensely studied and best understood mechanism, and also the most frequently used method to attenuate Notch signaling in pharmacological research and therapy. This is true despite the observation that the γ-Secretase complex targets approximately 140 different substrates [111] and is involved in the processing of many other proteins beside NOTCH receptors, although this may be the primary function with which this class of compounds is now associated. The initial target for the first GSI, however, was Alzheimer’s disease (AD), in which the γ-secretase complex is associated with the deposition of senile plaques, resulting from the aberrant deposition of amyloid precursor protein (APP) and the production of short amyloid-β peptides (Aβ) [112]. However, GSI have not proven beneficial for the treatment or prophylaxis against AD, and all clinical trials have failed. The first GSI to be used in clinical trials against cancer in the year 2006 was, therefore, a drug repurposed from AD research, MK-0752 [113]. The drug was only partially successful, and the clinical trial overall was a disappointment; MK-0752 was a non-specific, broad-band GSI and showed only limited efficacy in inhibiting T-ALL proliferation. Nevertheless, it resulted in limited disease regression in a small number of patients. Since 2006, many more clinical trials have been conducted with a spectrum of different GSI, most of which have failed, mainly due to high cytotoxicity and/or poor efficacy. Many initial clinical phase 1 studies, aiming to find the optimal and safe drug dosage for patients, have shown excessive toxicity in patients treated with GSI, mainly due to severe gastrointestinal adverse events, often resulting in the discontinuation of these trials. Many patients enrolled in these clinical trials suffered from persistent, severe diarrhea in a dose-dependent manner [114]. The basis for this toxicity remains incompletely understood. It is likely related to the existence of multiple known and validated γ-secretase substrates in addition to APP or NOTCH1, such as ephrinB2, ERBBP4, E-cadherin, and CD44 [115], which are widely expressed in human tissues, especially the gastrointestinal tract, and some of which may also be involved in the regulation of cell renewal or carcinogenesis [116,117]. As a result, the direct on-target inhibition of Notch signaling by GSI can result in severe gastrointestinal side effects, which can be persevered by additional, off-target effects. Notch signaling regulates the differentiation of progenitor cells and/or the maintenance of stem cell status in the human intestinal epithelium. Consequently, the blocking of Notch signaling by GSI can induce clinical, endoscopic, and histological abnormalities in the gastrointestinal tract, including secretory cell metaplasia of the intestinal epithelium which, in turn, leads to diarrhea [118]. Additionally, the inhibition of Notch signaling by GSI results in the loss of crypt base columnar (CBC) stem cells and an increase in secretory goblet cells’ population at the expense of proliferating cells. Consequently, γ-Secretase can result in altered homeostasis in the gastrointestinal system due to exposure to GSI, leading to rapid weight loss and even death, consistent with a failure of tissue replenishment and lack of nutrient absorption [119]. This has significantly limited the use of GSI for the treatment of cancer patients, forcing clinicians to utilize low, suboptimal, and intermittent dosing, thus decreasing the potential drug efficacy. Some of the toxicity observed in early clinical trials can also be alleviated by co-treatment with glucocorticoids. The problem was intrinsic to first-generation GSI, including drugs like AL101 and its follow-up development AL102 [120,121], a potent, orally administered γ-secretase inhibitor, but also Crenigacestat, and Nirogacestat. All of these drugs showed similar gastrointestinal toxicities, often coupled with anemia, fatigue, or thrombocytopenia [122,123]. Even today, there is still no strong evidence for their benefit in clinical use in anticancer therapy, which has been demonstrated in a series of recent clinical trials with Crenigacestat [123,124,125,126] and other GSI [127] (see Table 1 for an overview of the recent and ongoing clinical trials). This is combined with a frequent lack of adequate postoperative response. In addition, it has been shown that the actual effectiveness of some next-generation GSI, such as Semagacestat, in specifically blocking the function of the intended target γS may be much lower than expected [128]. However, there is still hope, and trials with GSI continue to this day, as illustrated in Table 1. Recently, Nirogacestat has shown a slightly better therapeutic outcome and better tolerance, and resulted in a significant increase in progression-free survival rate and lower side effects in patients with desmoid tumors [122].

Despite the disappointment of clinical trials, and due to the continuous high demand for useful tools in academic and preclinical research, a considerable number of small molecular weight compounds have been developed since, approximately, the year 2000, all of which inhibit γ-secretase and are now successfully used in laboratory practice. The GSI now represents a large and quite heterogeneous group of small molecular weight compounds, including peptide analogs and non-peptide small-molecule drugs, that share the capacity to inhibit the activity of the γ-secretase complex [129]. According to the National Library of Medicine database (National Institutes of Health; https://www.ncbi.nlm.nih.gov (accessed on 26 May 2023), there are currently 38 registered compounds described as γ-secretase modulators. The mechanism of GSI action is mostly related to the competitive binding and blocking of the active site of presenilin for its substrates. Alternatively, compounds may interfere in a non-competitive fashion with the structure and formation of the γ-secretase complex. GSI are commonly used as pan-Notch inhibitors in basic research, e.g., to investigate the role of Notch signaling in cancer development and progression. It is worth remembering that GSI also block all other, non-cancer-related, Notch-related processes and disorders. They are now commonly used as pan-Notch inhibitors in fundamental research related to the impact of Notch signaling in tissue development, cell fate decisions, differentiation, cell signaling, but also in stress response and various disorders [130,131,132,133]. One of the most commonly used GSI is the dipeptide DAPT, a presenilin inhibitor, which acts by binding to the catalytic pocket of PS, consequently blocking the internalization of the substrate and PS activation [134]. DAPT has been successfully used as a model compound (the “gold standard”) in many studies in which the inhibition of Notch signaling is suspected to have an anti-cancer effect. The use of DAPT includes studies on osteosarcoma [135], breast [136,137,138,139], ovarian [140], cervical [141,142], prostate [143], gastric [144,145], colorectal [146], and tongue [147] cancer, making it one of the most popular GSI used in cancer research.

In addition to DAPT, there are several extensively studied and tested GSI used to prevent NOTCH activation in cancer, including benzazepines such as Dibenzazepine (DBZ; YO-01027) [148,149,150], Crenigacestat (LY3039478) [151,152], LY411575 [153], and RO4929097 [154,155], sulfonamide derivatives such as Avagacestat (BMS-708163) [156,157], MK-0752 [158] and Venetoclax [159], or even natural compounds such as Hesperidin, a polyphenolic glycoside flavonoid that has shown anticancer potential in a colon cancer model [160]. In a broader sense, this further includes additional natural compounds such as Curcumin [161] or Evodiamine, an indole alkaloid that inhibits NOTCH3 signaling in a lung cancer model [162]. In most experimental studies, these compounds are used similarly to DAPT to inhibit γ-secretase activity and, as a consequence, Notch signaling [163,164]. However, some of the newer compounds may act with greater affinity and efficacy for γS, and can therefore be used at a significantly lower concentration, for example, as reported for LY411575 [165]. Additionally, the concomitant inhibition of Notch and tyrosine kinase receptor DDR1 (Discoidin Domain Receptor 1) signaling by LY411575 and 7rh, respectively, was demonstrated to trigger the blockade of cell growth and to induce apoptosis in Kras^G12V^-driven tumors via the modulation of MAPK signaling. Additionally, the combination of LY411575 and 7rh induced higher apoptosis than either agent alone [166]. This shows the potential of combining Notch inhibitors with other compounds to achieve the synthetic lethality phenomenon.

Several studies suggest the usefulness of GSI in anticancer treatment based on in vivo and in vitro cancer models. For example, Crenigacestat has demonstrated anticancer activity in a model of intrahepatic cholangiocarcinoma by inhibiting tumor cell growth, oncogenic activity of CAFs, and blood vessel formation [152,167]. Interestingly, in some cases, the use of GSI is associated with a simultaneous reduction in the cytotoxicity of other drugs and the promotion of cellular regeneration processes. The inhibition of γS by Dibenzazepine has been demonstrated to be protective against cisplatin-induced nephrotoxicity [168], spleen injuries [169], paracetamol-induced hepatotoxicity [170], and neomycin-induced cochleae [171] damage. This makes Dibenzazepine an interesting compound for research in adverse events and off-target effects, e.g., by reducing the non-specific cytotoxicity of anticancer drugs like the widely used cisplatin, along with the attenuation of Notch signaling.

Unfortunately, many GSIs are still considered to be of low efficacy and low specificity as anti-cancer drugs, coupled with a high risk of poor tolerance and severe side effects. However, their wide spectrum of activity against multiple targets seems, simultaneously, to be one of their greatest advantages and disadvantages in therapy. A reasonable and ethical solution might be to use GSI in a pre-selected cohort of patients with known Notch mutation status, which would only allow the inclusion of patients with confirmed hyperactive, oncogenic Notch signaling. However, this may be difficult, in practice; there are no accepted biomarkers that would indicate overall Notch pathway activity in cells and tissues and this is further complicated by the possibility that Notch pathway activity may be significantly affected by other genes than the Notch receptors themselves.

Another solution may be the use of natural organic compounds, usually characterized by lower toxicity and side effects. Curcumin is a natural polyphenol with pleiotropic effects on the functions and activity of transcription factors (STAT, NF-κB PPAR-γ, β-catechin), growth factors (VEGF, TGF-β1), protein kinases (MAPK, EGFR, JNK, IKK), and other proteins (Bcl-2, cMyc, cyclin D1) [172,173]. Additionally, the ability of Curcumin to inhibit γS has been demonstrated [161]. Despite data that show that Curcumin may act as a Notch signaling activator [174], or that it can reverse the effects of Notch inhibition [175] in normal (non-transformed) cells, research suggests its potent inhibitory effect on Notch signaling in cancer cells. The exposure of cells and tissues to Curcumin decreases Notch activation in T-ALL [176], thymic carcinoma [177], lung cancer [178], prostate cancer [179], or melanoma [180] cells.

Compounds like Curcumin are also known as “frequent hitters” that pop up in many pharmacological screens and have, therefore, been classified as “Pan-Assay INterference compoundS (PAINS)”. Other compounds that are candidates for PAINS are EGCG (epigallocatechin-3-gallate), Sulforaphane, Resveratrol, and Genistein. These are sometimes called “the big five” phytochemicals. All these compounds show remarkable polypharmacology and may bind to or interfere with many different substrates besides Notch, thus posing a high risk of causing false-positive assay signals. Nevertheless, they have in common that all of them have been associated with targeting cancer stem cells and EMT [181,182]. In addition to the Wnt/β-catenin, Hedgehog, PI3K/Akt/mTOR, and STAT3 pathways, the Notch pathway is another crucial molecular pathway that is prominently active in cancer stem cells [183]. The same pathways have been associated with EMT, relapse, dormancy, and increased tumor cell plasticity. Despite understandable doubt or outright criticism based on their poor target specificity, several recent clinical trials have demonstrated the therapeutic efficacy of these five phytochemicals. They have also been successfully combined with other cancer therapeutics [184] and in various types of cancer. Even if Notch signaling may be only one of several candidate pathways targeted by these “big five” phytochemicals, it may yet be worthwhile to seriously consider their capacity to target aggressive cancer stem cells, which are not affected by most chemotherapeutic drugs. Furthermore, it has been shown that derivatives of curcumin, so-called curcuminoids, can show much more specific effects on Notch pathway inhibition [176], suggesting that the pharmacophore of curcuminoids may, indeed, be promising for future medicinal chemistry and the development of more specific agents.

Less frequent are the pharmacological attempts to inhibit Notch signaling by blocking the S2 cleavage step, performed by ADAM proteases (ADAM-10, ADAM-17). For the same reasons as outlined for GSI, these compounds are also characterized by pleiotropic effects on multiple targets. They are not restricted to Notch signaling, but also inhibit, for example, matrix metalloproteinases like MMP2 and MMP9 [185]. The effective inhibition of ADAM-17, resulting in a reduction in Notch signaling by Marimastat (carboxamide), was observed in renal carcinoma [186] and by the compound ZLDI-8 in non-small-cell lung cancer [187] and colorectal cancer [188]. In addition, INCB7839 (Aderbasib), a dual-specific inhibitor targeting both ADAM10 and 17, has been tested in clinical trials against pediatric glioma. It appears, however, that this compound generally inhibits the shedding of many ligands required for cell–cell signaling, and has also been active, for example, against EGF receptor signaling [189,190]. In some cases, the results of inhibition of ADAM-17 seem to reflect the results of inhibition of GSI, i.e., inhibition of tumor cell growth and mobility as well as induction of apoptosis. However, the relatively small amount of reliability in this area makes it difficult to assess the usefulness of inhibiting ADAM proteases in the inhibition of Notch signaling and anticancer therapy.

## 5. NICD-Dependent Transcription in the Nucleus

The assembly and operation of the NICD/RPB-J/MAML transcription complex (a.k.a. NTC) is essential for the transcriptional initiation of Notch target genes (Figure 4). Several small molecular weight compounds have been developed that specifically modify the functionality of the complex. They act mainly by preventing individual subunits from binding to the complex. In most cases, this leads to the inhibition of the expression of genes dependent on Notch signaling. For example, the rhodanine ester compound IMR-1 (Mastermind Recruitment Inhibitor-1), and its metabolite IMR-1A, were demonstrated to interfere with the recruitment of MAML1 to RPB-J [191], consequently blocking the formation of the NTC and downstream gene expression in an esophageal adenocarcinoma model. Similarly, an α-helical peptide derived from mastermind-like 1 (SAHM1) acts as a dominant-negative MAML1 mimetic. Like IMR-1 and IMR-1A, SAHM1 attenuates Notch signaling by blocking the MAML/RPB-J interaction and its antitumor effects have been reported against bladder cancer [192] and acute myeloid leukemia [193]. However, SAHM1 showed no significant effects on B-cell acute lymphoblastic leukemia (B-ALL) [194]. The synthetic molecule CB-103 blocks the functional interaction between RBP-J and NICD, showing antitumor activity against T-ALL cells, TNBC xenografts in mouse models [195], and T-ALL in clinical studies [196]. Additionally, CB-103 has shown the ability to inhibit Notch signaling in other cancer models, including clear-cell renal cell carcinoma [197]. Presently, CB-103 is in at least one Phase 2 clinical study (Study of CB-103 in Adult Patients with Advanced or Metastatic Solid Tumours and Haematological Malignancies; ClinicalTrials.gov Identifier: NCT03422679, see Table 1).

The downregulation of NOTCH receptors has also been shown with pan-HDAC inhibitors such as Givinostat and Trichostatin A in T-ALL. More detailed analysis showed that this effect is related to the inhibition of HDAC6, which can also be achieved by using more specific inhibitors, such as Tubacin [198]. HDACs, along with RBP-J and other co-repressor proteins, are components of the inactive transcription complex, which blocks the expression of Notch-targeted genes. Additionally, the inhibition of HDAC6 has been shown to lead to the disruption of NOTCH3 receptor transport. This leads to the increased accumulation and degradation of NOTCH3 in lysosomes, which results in the attenuation of Notch signaling [198]. This shows a direct relationship between the functions of the NICD-dependent nuclear transcription complex and membrane trafficking of NOTCH receptors.

## 6. Small Molecular Weight Compounds with Pleiotropic Functions in Relation to Notch Signaling

The activation of Notch signaling has shown some beneficial outcomes in studies of certain types of cancer. The possibility of the activation of genes dependent on Notch signaling, because of the modulation of the previously described Notch ternary complex, was confirmed. Very often, however, the compounds used for this purpose show a context-dependent effect.

RIN-1 (RBPJ Inhibitor-1; 2-(2-Fluorophenoxy)-4-(1-methyl-1H-pyrazol-5-yl) benzamide) is an inhibitor that blocks the functional interaction of RBP-J with SHARP (also known as SMRT/HDAC1-associated repressor protein or MINT-Msx2-interacting nuclear target) [199]. SHARP is a scaffold protein that functions in multiple ways: (1) SHARP acts as an NTC co-repressor by binding NCoR1/2 (Nuclear receptor co-repressor 1 and 2) and HDAC, consequently inhibiting transcription. In addition, (2) SHARP is also an NTC co-activator that acts by recruiting KMT2D lysine methyltransferases [200,201] to the complex. Despite the bimodal function of SHARP, studies suggest that blocking SHARP/RPB-J interaction by RIN-1 leads to an increase in the expression of genes dependent on Notch signaling [199]. Therefore, RIN-1 can be generally considered an activator of Notch signaling.

Valproic acid (VPA), an HDAC inhibitor, is a recognized stimulator of Notch signaling and exhibits anticancer properties in various cancer types, such as cervical cancer (resulting in cell cycle arrest, proliferation suppression, and apoptosis induction) [202] and human pancreatic carcinoid (as above) [203]. On the other hand, it has been shown that VPA can also attenuate Notch signaling, as was the case with chronic myelogenous leukemia K562 cells [204], and hepatocellular carcinoma HepG2 [205] cells. In addition to HDAC inhibition (selectivity to HDAC6 and HDAC8), VPA works by enhancing γ-aminobutyric acid (GABA) signaling and inhibiting calcium and sodium channels [206], making it difficult to use as a selective inhibitor of Notch signaling. Moreover, a significant limitation of the use of VPA may be its relatively low efficacy: concentrations usually exceeding 4 mM are necessary to obtain a measurable modulation of Notch signaling.

Another widely used compound that can both activate or attenuate Notch signaling is Resveratrol, one of the “Big Five” natural compounds. Interestingly, antitumor activity was reported for both modes of action; Resveratrol (and other polyphenols) are well-known activators of Notch signaling in normal, non-transformed cells, including endothelial cells [207], but also some types of cancer, including osteosarcoma [208], neuroendocrine carcinoid [209], and thyroid cancer [210,211]. However, the currently available data on the putative activation of NOTCH signaling in cancer by Resveratrol are ambiguous and suggest that the compound may suppress Notch signaling. For example, an anticancer effect accompanied by the downregulation of proteins, such as NOTCH1, Jagged1, DLL4, and HES5, by Resveratrol was observed in MDA-MB-231 breast cancer cells [212]. In OVCAR-3 ovarian cancer cells, the downregulation of NOTCH-2 and HES1 was observed [213] while, in the cell lines A2780 and SKOV3, a reduction in full and cleaved NOTCH1 was observed [214]. In some cases, the net effect of Resveratrol does not allow the determination of its mechanism of action as an activator/inhibitor of Notch signaling, as was the case with Glioblastoma multiforme (GBM) cell lines U87MG and T98G, in which a simultaneous decrease in NOTCH1 and NICD expression and an increase in HES1 expression was observed [155]. Resveratrol is also known as a pan-inhibitor of histone deacetylases in cancer cells [215], which may explain its potential to modulate Notch signaling. In addition, Resveratrol interacts with elements of the various signaling pathways, such as β-catenin, Smad2/3, IKK, AKT, PI3K, STAT3, and HIF-1α, which are closely related to canonical and non-canonical Notch signaling modulation [216]. Consequently, the result of its action is, again, pleiotropic and often indirect, similar to that described for Curcumin, and it is difficult to determine the efficacy and specificity of these compounds in relation to Notch signaling.

Another broad cellular mechanism that has been exploited to target NOTCH1 protein levels within cancer cells is the use of proteasome inhibitors, such as Bortezomib. Bortezomib effectively represses the transcription of NOTCH1, thus blocking the activation of Notch downstream effectors including HES1, GATA3, or RUNX3, and resulting, for example, in the downregulation and dissociation of the major transcriptional activator Sp1 from the NOTCH1 promoter. This causes significant cytotoxicity in T-ALL cells [217] which is partly overcome by the overexpression of the NOTCH1 intracellular domain (NICD). Furthermore, a combination of Bortezomib with the histone deacetylase inhibitor romidepsin further increased the cytotoxic effects [218], at least in a mouse model of leukemia. Interestingly, Bortezomib has been shown to increase the expression of the Notch-dependent genes downstream in a γ-secretase-independent manner in tumor-draining CD8+ T cells [219]. This demonstrates the significant context-dependent influence of proteasome activity on Notch signaling. It has also been debated whether part of the effects of GSI in breast cancers may be related to targeting the proteasome [220], effects that have also been observed in neuroblastoma. It remains unclear if the effects of proteasome inhibitors against cancers may partly relate to the blocking of Notch signaling, and vice versa, if the anti-cancer effects of GSI may, at least partly, stem from the inhibition of the proteasome. It should also be mentioned that, Fostamatinib is functionally related to proteasome activity via NUMB activity, which blocks calcium-/calmodulin-dependent protein kinase CAMK1. CAMK1 regulates NUMB-mediated endocytosis by the phosphorylation of NUMB on the Serine residue 276 and 295; interfering with this phosphorylation effectively blocks the nuclear import of the NICD (for reference, see GenBank, https://go.drugbank.com/drugs/DB12010 (accessed on 26 May 2023).

Last but not least, an interesting concept has been introduced that specifically targets mutated tumor suppressor genes with nonsense mutations. Apart from TP53, NOTCH1 and FAT1 also show high frequencies of nonsense mutations, especially in HNSCC. The treatment of NOTCH-mutated HNSCC cell lines with the compound PTC-124 (Ataluren) was demonstrated to re-express the functional NOTCH1 protein and induce the expression levels of Notch-regulated genes such as HES1, HES5, and AJUBA, thus effectively reverting the loss-of-function mutations found in the tumor cells [221]. PTC-124 selectively induces the ribosomal readthrough of premature but not normal termination (stop) codons. Mechanistically, PTC-124 acts by binding to rRNA, which inhibits the ribosomal activity to mRNA termination. Consequently, PTC-124 mostly allows the “bypassing” of UGA nonsense mutations in the mRNA. This makes the synthesizing of a full-length protein possible with a changed amino acid residue at the mutation site [222,223,224,225]. PTC-124 was originally developed for the treatment of cystic fibrosis, where it was also able to functionally rescue some nonsense mutations in the CFTR transporter gene [226]. Its application in cancer is a classic example of drug repurposing. Additional examples with relevance for Notch signaling have been described in the recent literature [159], not only identifying the Bcl-2 inhibitor Venetoclax as a γ-secretase inhibitor, but also indicating its superior efficacy over other “standard” GSI such as RO4929097. Another example is the identification of the FDA-approved antibiotic Fidaxomicin as a potential RBP-J inhibitor [227]. Fidaxamicin was subsequently shown to block RBP-J-dependent transcription and thereby inhibit the Notch-dependent growth of TNBC. Similarly, the lipid-lowering agent Lomitapide was identified as a potent compound targeting proteolytic enzymes, most prominently ADAM17 and the γ-secretase complex [228]. It showed remarkable proliferation-blocking, antitumor properties, especially against TNBC cells and cell lines. Lomitapide also induced significant apoptosis and blocked cell cycle progression in TNBC cells. Another concept that may be relevant here is the use of synthetic lethality to target cancer cells that have mutated (loss-of-function) Notch receptors. This was suggested over 10 years ago [229], but only recently has the concept been experimentally demonstrated in oral cancers. A broader acceptance of this strategy might be very useful in developing more successful combination therapies, either of targeted compounds specifically targeting tumors with functional loss of Notch signaling, or vice versa, using Notch inhibitors to target cancer cells that have losses in other tumor suppressor genes. A thorough investigation of patient-specific pathway activity is the condition for this concept to work in personalized cancer medicine, but the initial steps have been taken and Notch inhibitors may be “on board”.

Currently, selective activators of Notch signaling are much more difficult to obtain (and validate) than inhibitors. This leads to several inconveniences when planning experiments, including the selection of appropriate cancer cell lines with clearly traceable Notch activity, as well as assessing the likely interaction of the tested compounds with other signaling pathways.

A summary of the clinical trials for small-molecule compounds targeting Notch signaling is presented in Table 1.

## 7. Conclusions

The development of research on small-molecule modulators of Notch signaling offers new perspectives for a better understanding of the processes behind cancer initiation and progression, including processes that lead to poor patient survival, such as relapse, acquired drug resistance, increased aggressive behavior, and metastasis. Currently, Notch small-molecule modulators are successfully used and are relatively convenient and reliable in laboratory practice, despite the notorious question related to their specificity. Nevertheless, they have advantages over recombinant monoclonal antibodies that either target Notch receptors or ligands, such as Brontictuzumab, a therapeutic antibody targeting, specifically, NOTCH1 [230,231,232]. The realistic potential of Notch pathway modulators (inhibitors and a few activators) in basic research, applied medicine, and eventually, in clinical trials is still under consideration. Despite significant recent progress in this field, especially in the identification of targets besides the γ-secretase complex and the proteolytic processing of the Notch receptors themselves, the biggest problem appears to be the low selectivity of most compounds. This can be alleviated by increasingly targeting the Notch receptor and addressing the need for the careful stratification of patient cohorts in order to identify those candidates that may target downstream signaling events, such as the NICD/RBP-J/MAML1 complex and transcriptional activity, instead of directly targeting the Notch receptors themselves. In clinical treatment, obtaining appropriate Notch modulators that are capable of interfering with Notch signaling without excessive off-target and non-specific side effects represents an urgent, unmet clinical need, but also a near-term, realistic goal for medicinal chemistry in collaboration with translational cancer research and personalized medicine.

## Figures and Tables

**Figure 1 cancers-15-04563-f001:**
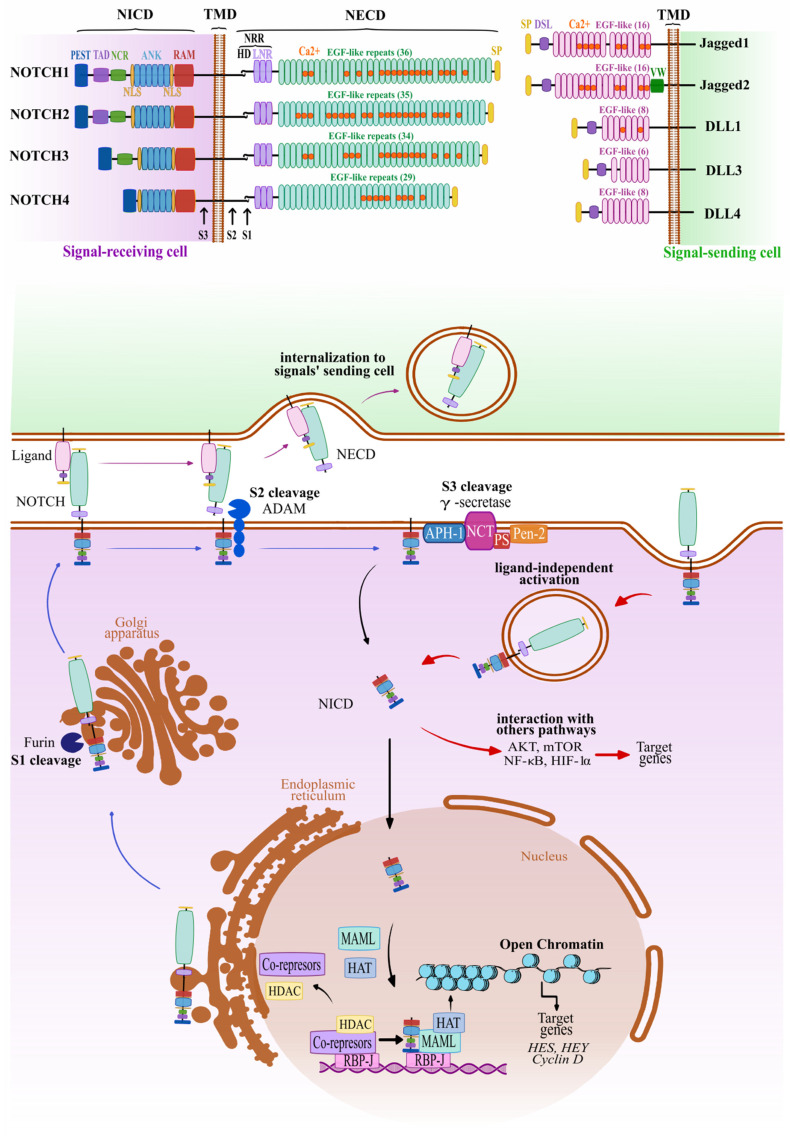
Canonical and non-canonical Notch signaling pathway. A schematic figure of the structure of the receptors (NOTCH1-4) and their ligands (DLL-1/3/4 and JAG1/2). After synthesis, the NOTCH receptors mature in a stepwise process and are simultaneously transported through the endoplasmic reticulum and Golgi structures to the plasma membrane. Receptor–ligand interactions generate a mechanical pulling force that allows ADAM proteases and the γ-secretase complex to cleave the receptor (at the S2 and S3 cleave site, respectively), followed by the release of the intracellular domain (NICD) to the cytoplasm of the signal-receiving cell. Simultaneously, the extracellular part of the NOTCH (NECD) and the ligand are internalized by the signal-sending cell. Intracellular release of NICD can also occur without ligand activation via NOTCH internalization and γ-secretase activity. The NICD is translocated into the nucleus where it participates in the formation of the Notch ternary complex (NTC), activating the expression of target genes. In non-canonical Notch signaling, the NICD interacts with and modulates other signaling pathways to express target genes without the mediation of the NTC.

**Figure 2 cancers-15-04563-f002:**
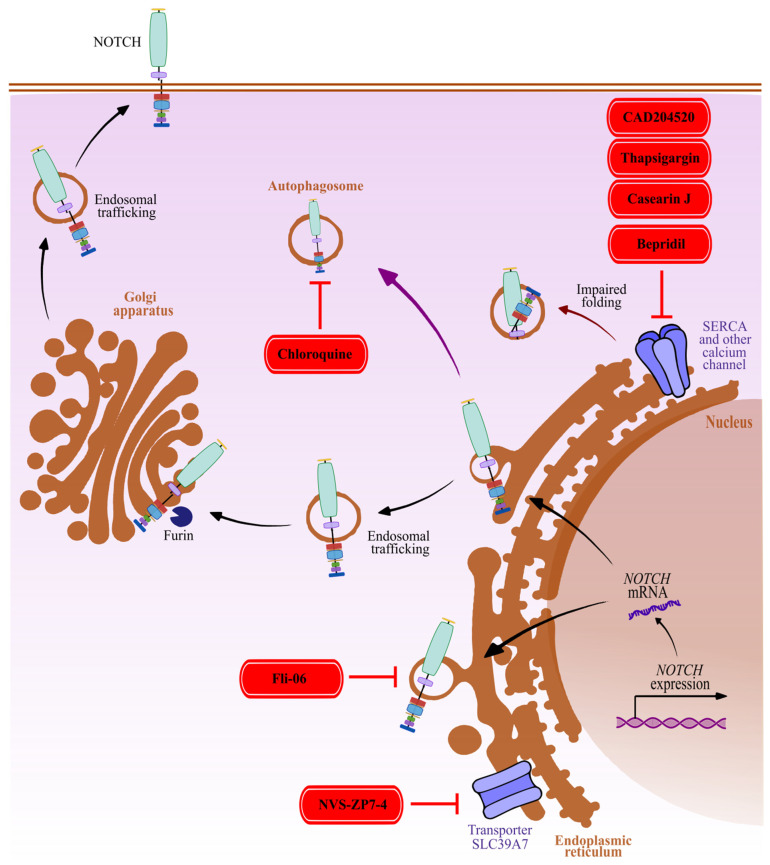
Modulation of Notch signaling by interfering with membrane trafficking of NOTCH receptors. Inhibition of the membrane ion transporters SLC39A7 (NVS-ZP7-4) and SERCA (Thapsigargin, CAD204520, Casearin J) disrupts Zn^2+^ and Ca^2+^ ion efflux between endoplasmic reticulum/Golgi structures and the cytoplasm, resulting in impaired protein folding, arrested maturation, and a block of intracellular NOTCH protein transport. Attenuation of receptor exposure to the membrane can also be achieved by compounds blocking NOTCH secretion prior to the exit from the endoplasmic reticulum (FLI-06), or leading to the accumulation of NOTCH receptors in autophagosomes (Chloroquine, more details in the text). Red: attenuation of Notch signaling (NVS-ZP7-4, Thapsigargin, CAD204520, Casearin J, Bepridil, FLI-06, Chloroquine).

**Figure 3 cancers-15-04563-f003:**
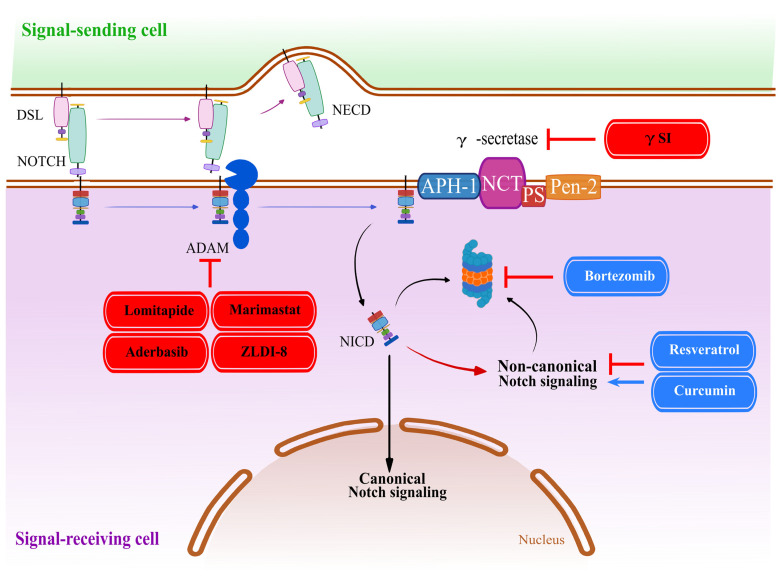
Modulation of NICD release. ADAM protease inhibitors (Lomitapide, Aderbasib, Marimastat, ZLDI-8) block NOTCH S2 cleavage and, consequently, further activation of the NOTCH receptor. The use of γ-secretase inhibitors (γSI; GSI) prevents S3 cleavage and subsequent NICD release. The NICD can be translocated to the nucleus where it is involved in canonical Notch signaling or interacts with other signaling pathways as a component of non-canonical Notch signaling. These events are influenced by proteasome activity (inhibited by Bortezomib) and a number of broad-spectrum substances collectively named as Pan-Assay INterference compoundS (PAINS) such as Resveratrol and Curcumin. Red: attenuation of Notch signaling; Blue: pleiotropic effect on Notch signaling.

**Figure 4 cancers-15-04563-f004:**
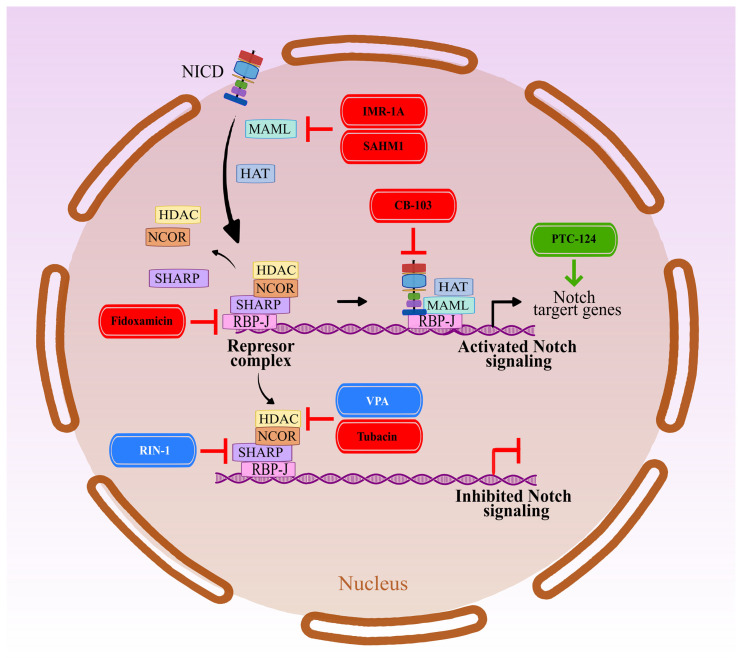
Modulation of Notch signaling by targeting the NICD-dependent transcription complex. Attenuation of Notch signaling can be achieved by blocking ternary complex formation by inhibiting RBP-J activity (Fidaxamicin) or preventing the binding of MAML to NICD/RPB-J (IMR-1A, SAHM1) or NICD to RBP-J (CB-103). RIN-1 blocks the binding of the SHARP co-repressor to RBP-j, preventing the function of the repressor complex. HDAC inhibitors (Givinostat, Trichostatin, Tubacin, Valproic acid) block the ability of HDACs to deacetylate histones, contributing to the maintenance of an open DNA structure and expression of genes. Finally, some compounds, such as PTC-124, can re-express functional NOTCH1 and induce Notch-regulated gene expression levels. Red: attenuation of Notch signaling (IMR-1, SAHM1, CB-103, Tubacin), Blue: pleiotropic effect on Notch signaling (RIN-1, VPA), Green: activation of Notch signaling.

**Table 1 cancers-15-04563-t001:** Clinical trials of small molecular compounds targeting Notch signaling.

Study Title	NCT Number	Status	Conditions	Interventions
A Study Of AL101 in Patients with Adenoid Cystic Carcinoma (ACC) Bearing Activating Notch Mutations	NCT03691207	Unknown status	Adenoid Cystic Carcinoma	AL101
AL101 Before Surgery for the Treatment of Notch-Activated Adenoid Cystic Cancer	NCT04973683	Recruiting	Adenoid Cystic Carcinoma	AL101
A Study of AL101 Monotherapy in Patients with Notch-Activated Triple-Negative Breast Cancer	NCT04461600	Active, not recruiting	Triple-Negative Breast Cancer	AL101
A Study of AL102 in Patients with Progressing Desmoid Tumors	NCT04871282	Recruiting	Desmoid Tumor	AL102
Single-arm Study with Bimiralisib in Patients With HNSCC Harboring NOTCH1 Loss of Function Mutations	NCT03740100	Terminated	HNSCC	Bimiralisib (PI3Kinase inhibitor)
Study to Evaluate the Safety and Tolerability of Weekly Intravenous (IV) Doses of BMS-906024 in Subjects with Acute T-cell Lymphoblastic Leukemia or T-cell Lymphoblastic Lymphoma	NCT01363817	Completed	Acute T-cel Lymphoblastic Leukemia, Precursor T-Cell Leukemia	BMS-906024 (γ-secretase inhibitor)
Study to Evaluate the Safety and Tolerability of IV Doses of BMS-906024 in Subjects with Advanced or Metastatic Solid Tumors	NCT01292655	Completed	Cancer	BMS-906024
Study to Evaluate Safety and Tolerability of BMS-906024 in Combination with Chemotherapy	NCT01653470	Completed	Solid cancers	BMS-906024, Paclitaxel, 5-Fluorouracil (5FU), Carboplatin
Phase I Ascending Multiple-Dose Study of BMS-986115 in Subjects with Advanced Solid Tumors	NCT01986218	Terminated	Various Advanced Cancers	BMS-986115 (γ-secretase inhibitor)
Compassionate Use of Brontictuzumab for Adenoid Cystic Carcinoma (ACC)	NCT02662608	Completed	Adenoid Cystic Carcinoma	Brontictuzumab (monoclonal antibody that targets Notch1)
CB-103 Plus NSAI in Luminal Advanced Breast Cancer	NCT04714619	Active, not recruiting	Advanced Breast Cancer	CB-103, NSAIDs
Study of CB-103 in Adult Patients with Advanced or Metastatic Solid Tumors and Hematological Malignancies	NCT03422679	Terminated	Breast Cancer, Colorectal, ACC, Osteosarcoma, HCC	CB-103
A Phase 1/2 Study CB-103 with or without Venetoclax in Patients with NOTCH ACC	NCT05774899	Recruiting	Adenoid Cystic Carcinoma	CB-103, Venetoclax, Lenvatinib
Study in Patients with Advanced Cancers Associated with Expression of DLL3	NCT04471727	Recruiting	Small-cell Lung Cancer	HPN328 (a DLL3-targeting T-cell engager), Atezolizumab
A Study of LY3039478 in Japanese Participants with Advanced Solid Tumors	NCT02836600	Active, not recruiting	Advanced Solid Tumor	Crenigacestat/LY3039478
A Study of LY3039478 in Participants with Advanced or Metastatic Solid Tumors	NCT02784795	Completed	Solid Tumors: Breast Cancer, Colon Cancer	Crenigacestat/LY3039478, Taladegib, Abemaciclib, Cisplatin, Gemcitabine, Carboplatin
Notch Inhibitor in Advanced Cancer	NCT01158404	Completed	Advanced Cancer	LY900009 (γ-secretase inhibitor)
A Phase 1 Study to Evaluate the Safety, Tolerability, and Pharmacokinetics of MEDI0639 in Advanced Solid Tumors	NCT01577745	Completed	Solid Tumors	MEDI0639 (DLL4-targeting antibody)
A Notch Signaling Pathway Inhibitor for Patients with Advanced Breast Cancer (0752-014)	NCT00106145	Completed	Advanced Breast Cancer, other Solid Tumors	MK-0752
A Notch Signaling Pathway Inhibitor for Patients with T-cell Acute Lymphoblastic Leukemia/Lymphoma (ALL) (0752-013)	NCT00100152	Terminated	Myelogenous Leukemia, Chronic Lymphocytic Leukemia, Lymphoblastic Acute T-cell Leukemia	MK-0752
Phase I/II Study of MK-0752 Followed by Docetaxel in Advanced or Metastatic Breast Cancer	NCT00645333	Completed	Metastatic Breast Cancer	MK-0752, Docetaxel, Pegfilgrastim
MK-0752 and Gemcitabine Hydrochloride in Treating Patients with Stage III and IV Pancreatic Cancer that Cannot be Removed by Surgery	NCT01098344	Completed	Pancreatic Cancer	MK-0752, gemcitabine
Nirogacestat for Adults with Desmoid Tumor/Aggressive Fibromatosis (DT/AF)	NCT03785964	Active, not recruiting	Desmoid Tumor, Aggressive Fibromatosis	Nirogacestat
Individual Patient Compassionate Use of Nirogacestat	NCT05041036	Available	Desmoid Tumors, NOTCH Gene Mutation Positive Tumors	Nirogacestat
γ-secretase inhibitor PF-03084014 in Treating Patients with AIDS-Associated Kaposi Sarcoma	NCT02137564	Withdrawn	AIDS-related Kaposi Sarcoma	PF-03084014
Biomarker Research Study for PF-03084014 in Chemo-resistant Triple-negative Breast Cancer	NCT02338531	Withdrawn	Breast Cancer	PF-03084014
A Study Evaluating The PF-03084014 in Combination with Docetaxel in Patients with Advanced Breast Cancer	NCT01876251	Terminated	Breast Cancer Metastatic	PF-03084014
Phase II Trial of the γ-Secretase Inhibitor PF-03084014 in Adults With Desmoid Tumors/Aggressive Fibromatosis	NCT01981551	Active, not recruiting	Desmoid Tumors, Aggressive Fibromatosis	PF-03084014
A Trial in Patients with Advanced Cancer and Leukemia	NCT00878189	Completed	Advanced Cancer And Leukemia	PF-03084014
A Study Evaluating PF-03084014 in Patients with Advanced Breast Cancer with Or without Notch Alterations	NCT02299635	Terminated	Triple-Negative Breast Neoplasms	PF-03084014
γ-secretase inhibitor RO4929097 in Previously Treated Metastatic Pancreas Cancer	NCT01232829	Completed	Adenocarcinoma of the Pancreas, Recurrent Pancreatic Cancer	RO4929097
RO4929097 Before Surgery in Treating Patients with Pancreatic Cancer	NCT01192763	Terminated	Adenocarcinoma of the Pancreas, Stage IA, IB Pancreatic Cancer	RO4929097
Vismodegib and γ-secretase/Notch Signaling Pathway Inhibitor RO4929097 in Treating Patients with Advanced or Metastatic Sarcoma	NCT01154452	Completed	Adult Alveolar Soft Part Sarcoma, Angiosarcoma, Desmoplastic Small Round Cell Tumor	RO4929097
RO4929097 in Treating Patients With Recurrent Invasive Gliomas	NCT01269411	Terminated	Adult Anaplastic Oligodendroglioma, Brain Stem Glioma, Giant Cell Glioblastoma	RO4929097
γ-secretase/Notch Signaling Pathway Inhibitor RO4929097 in Treating Patients with Recurrent or Progressive Glioblastoma	NCT01122901	Terminated	Adult Giant Cell Glioblastoma, Gliosarcoma	RO4929097
γ-secretase inhibitor RO4929097 in Treating Young Patients with Relapsed or Refractory Solid Tumors, CNS Tumors, Lymphoma, or T-Cell Leukemia	NCT01088763	Terminated	Childhood Atypical Teratoid/Rhabdoid Tumor, Choriocarcinoma, Germinoma	RO4929097
A Study of RO4929097 in Patients with Advanced Renal Cell Carcinoma that Have Failed Vascular Endothelial Growth Factor (VEGF)/Vascular Endothelial Growth Factor Receptor (VEGFR) Therapy	NCT01141569	Completed	Clear Cell Renal Cell Carcinoma, recurrent, and Stage IV Renal Cell Cancer	RO4929097
γ-secretase/Notch Signaling Pathway Inhibitor RO4929097 in Treating Patients with Advanced, Metastatic, or Recurrent Triple-Negative Invasive Breast Cancer	NCT01151449	Terminated	Triple-Negative Breast Cancer	RO4929097
γ-secretase/Notch Signaling Pathway Inhibitor RO4929097 in Treating Patients with Stage IV Melanoma	NCT01120275	Terminated	Malignant Melanoma	RO4929097
RO4929097 in Treating Patients with Metastatic Colorectal Cancer	NCT01116687	Completed	Recurrent Colon, Rectal Cancer, Stage IV Colon Cancer	RO4929097
RO4929097 in Treating Patients with Recurrent and/or Metastatic Epithelial Ovarian Cancer, Fallopian Tube Cancer, or Primary Peritoneal Cancer	NCT01175343	Completed	Fallopian Tube Carcinoma, Ovarian Carcinoma, Primary Peritoneal Carcinoma	RO4929097
RO4929097 in Treating Patients with Advanced Non-Small Cell Lung Cancer who have Recently Completed Treatment with Front-Line Chemotherapy	NCT01193868	Terminated	Recurrent Non-Small Cell Lung Cancer, Stage IIIB, Stage IV	RO4929097
RO4929097 in Treating Patients with Stage IIIB, Stage IIIC, or Stage IV Melanoma that Can be Removed by Surgery	NCT01216787	Withdrawn	Melanoma Stage IIIB, IIIC, IV	RO4929097
γ-secretase inhibitor RO4929097 in Treating Patients with Metastatic or Unresectable Solid Malignancies	NCT01096355	Completed	Unspecified Adult Solid Tumor	RO4929097
RO4929097 After Autologous Stem Cell Transplant in Treating Patients with Multiple Myeloma	NCT01251172	Withdrawn	Plasma Cell Myeloma Stage I, II, III	RO4929097, Autologous Hematopoietic Stem Cell Transplantation
RO4929097 and Bevacizumab in Treating Patients with Progressive or Recurrent Malignant Glioma	NCT01189240	Terminated	Adult Anaplastic Astrocytoma, Oligodendroglioma, Giant Cell Glioblastoma	RO4929097, bevacizumab
Combination Chemotherapy and Bevacizumab with or without RO4929097 in Treating Patients with Metastatic Colorectal Cancer	NCT01270438	Withdrawn	Adenocarcinoma of the Colon, Rectum	RO4929097, bevacizumab, FOLFOX regimen,
Bicalutamide and RO4929097 in Treating Patients with Previously Treated Prostate Cancer	NCT01200810	Terminated	Adenocarcinoma of the Prostate	RO4929097, bicalutamide
RO4929097 and Capecitabine in Treating Patients with Refractory Solid Tumors	NCT01158274	Completed	Adult Grade III Lymphomatoid Granulomatosis, Adult Nasal Type Extranodal NK/T-cell Lymphoma, AIDS-related Diffuse Large Cell Lymphoma	RO4929097, capecitabine
γ-secretase/Notch Signaling Pathway Inhibitor RO4929097, Paclitaxel, and Carboplatin Before Surgery in Treating Patients with Stage II or Stage III Triple-Negative Breast Cancer	NCT01238133	Terminated	Triple-Negative Breast Cancer	RO4929097, Carboplatin
Phase I Study of Cetuximab with RO4929097 in Metastatic Colorectal Cancer	NCT01198535	Terminated	Colon Mucinous Adenocarcinoma, Colon Signet Ring Cell Adenocarcinoma, Rectal Mucinous Adenocarcinoma	RO4929097, Cetuximab
γ-secretase/Notch Signaling Pathway Inhibitor RO4929097 in Combination with Cisplatin, Vinblastine, and Temozolomide in Treating Patients with Recurrent or Metastatic Melanoma	NCT01196416	Completed	Recurrent Melanoma, Stage IV Skin Melanoma	RO4929097, Cisplatin
RO4929097 in Children with Relapsed/Refractory Solid or CNS Tumors, Lymphoma, or T-Cell Leukemia	NCT01236586	Withdrawn	Lymphoma	RO4929097, Dexamethasone
RO4929097 and Erlotinib Hydrochloride in Treating Patients with Stage IV or Recurrent Non-Small Cell Lung Cancer	NCT01193881	Terminated	Recurrent Non-Small Cell Lung Carcinoma, Stage IV Non-Small Cell Lung Cancer	RO4929097, Erlotinib Hydrochloride
γ-secretase Inhibitor RO4929097 and Gemcitabine Hydrochloride in Treating Patients with Advanced Solid Tumors	NCT01145456	Completed	Adenocarcinoma of the Pancreas, Recurrent Pancreatic Cancer, Stage III	RO4929097, gemcitabine
γ-secretase/Notch Signaling Pathway Inhibitor RO4929097 in Treating Patients with Advanced Solid Tumors	NCT01218620	Completed	Adult Solid Neoplasm	RO4929097, Ketoconazole, Rifampicin
RO4929097 and Letrozole in Treating Post-Menopausal Women with Hormone-Receptor-Positive Stage II or Stage III Breast Cancer	NCT01208441	Terminated	Estrogen-Receptor-positive and Progesterone-Receptor-positive Breast cancer, HER2-negative Breast Cancer	RO4929097, letrozole
γ-secretase/Notch Signaling Pathway Inhibitor RO4929097 and Temsirolimus in Treating Patients with Advanced Solid Tumors	NCT01198184	Completed	Endometrial Papillary Serous Carcinoma, Recurrent Endometrial Carcinoma, Recurrent Renal Cell Cancer	RO4929097, temsirolimus
RO4929097 and Vismodegib in Treating Patients with Breast Cancer that is Metastatic or Cannot be Removed by Surgery	NCT01071564	Terminated	Triple-Negative Breast Cancer	RO4929097, Vismodegib
RO4929097 and Whole-Brain Radiation Therapy or Stereotactic Radiosurgery in Treating Patients with Brain Metastases From Breast Cancer	NCT01217411	Terminated	Triple-Negative Breast Cancer	RO4929097, Whole-brain radiation therapy (WBRT), Stereotactic radiosurgery (SRS)
RO4929097, Temozolomide, and Radiation Therapy in Treating Patients with Newly Diagnosed Malignant Glioma	NCT01119599	Completed	Acoustic Schwannoma, Anaplastic Astrocytoma, Anaplastic Meningioma	RO4929097, 3-Dimensional Conformal and Intensity-Modulated Radiation Therapy
A Phase 1b/2 Study of OMP-59R5 (Tarextumab) in Combination with Nab-Paclitaxel and Gemcitabine in Subjects with Previously Untreated Stage IV Pancreatic Cancer	NCT01647828	Completed	Pancreatic Cancer Stage IV	Tarextumab (OMP-59R5, a NOCTH2/3 targeting antibody), Gemcitabine
A Study of Tarlatamab in Participants with Neuroendocrine Prostate Cancer	NCT04702737	Active, not recruiting	Neuroendocrine Prostate Cancer	Tarlatamab (a DLL3-targeting T-cell engager)

## Data Availability

The data presented in this study are available in the presented article. Any additional data related to this study are available on request from the corresponding author.

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
