# Peer review of "Modulation of Notch Signaling by Small-Molecular Compounds and Its Potential in Anticancer Studies"

_cancers, 2023, doi:10.3390/cancers15184563_

Round 1

Reviewer 1 Report

The manuscript “Modulation of Notch Signaling by Small-Molecular Compounds and Its Potential in Anticancer Studies” by Czerwonka et al. submitted for review in Cancers describes in an interesting update using well-illustrated Figures how the development of research on small-molecule modulators of Notch signaling offers new perspectives for a better understanding of cancer and a strong opportunity in translational cancer research. This Ms is well written and extents with a different angle some aspects of a precedent review in the field that was published in 2021 and written by two of the authors.

This Ms might benefit to be published in Cancers with minor modifications that are suggested below:

-Although the Ms illustrates clearly in the different paragraphs (3 to 6) at which level in the Notch signalling pathway the different compounds act and can modulate it specifically in different types of cancers, I suggest to make efforts between paragraphs 1, which is interesting and well-written in itself, and 2 in order to avoid redundant information that appears along the text concerning notably Numb, Ajuba and MAML1.  

-In addition, I suggest to invert #1 and #2 and to shorten #1 because so many reviews had described the pathway already and authors can refer to some of them in light of their Fig.1. This might give more focus on the present #1.

-Finally, inverting L142-147 from present #1with L255-267 might be more appropriate regarding the aspects developed in each paragraph.

Author Response

Dear Reviewer,

We greatly appreciate your comments and suggestions on the manuscript. These comments are very helpful to improve our work. The manuscript has been modified in line with the Reviewers’ suggestions. We hope that the revised version of the manuscript will reach the Reviewers' expectations and make the manuscript appropriate to be published in Cancers.

The changes introduced in manuscript and the list of responses to the Reviewer’s comments are listed below. The proposed changes are marked in the main text in review mode.

Open Review 1

Comments and Suggestions for Authors:

The manuscript “Modulation of Notch Signaling by Small-Molecular Compounds and Its Potential in Anticancer Studies” by Czerwonka et al. submitted for review in Cancers describes in an interesting update using well-illustrated Figures how the development of research on small-molecule modulators of Notch signaling offers new perspectives for a better understanding of cancer and a strong opportunity in translational cancer research. This Ms is well written and extents with a different angle some aspects of a precedent review in the field that was published in 2021 and written by two of the authors.

This Ms might benefit to be published in Cancers with minor modifications that are suggested below:

Question: -Although the Ms illustrates clearly in the different paragraphs (3 to 6) at which level in the Notch signalling pathway the different compounds act and can modulate it specifically in different types of cancers, I suggest to make efforts between paragraphs 1, which is interesting and well-written in itself, and 2 in order to avoid redundant information that appears along the text concerning notably Numb, Ajuba and MAML1.  

-In addition, I suggest to invert #1 and #2 and to shorten #1 because so many reviews had described the pathway already and authors can refer to some of them in light of their Fig.1. This might give more focus on the present #1.

Answer: The order of chapters 1 and 2 in the manuscript has been reversed. Additionally, to avoid redundant information about Numb, Ajuba and MAML1, we decided to remove part of text from chapters (lines: 89-113). Current chapter 1 (Canonical and non-canonical Notch pathway) has been shortened.

Question: -Finally, inverting L142-147 from present #1with L255-267 might be more appropriate regarding the aspects developed in each paragraph.

Answer: Lines L142-147 and L255-267 have been relocated to the proper chapters.

Reviewer 2 Report

While Notch acts as tumor suppressor in squamous cell carcinoma in head and neck cancer (HNSCC) as well as in CSCC (cutaneous squamous cell carcinoma), the role of Notch signaling in squamous cell carcinoma of the lung remains to be more thoroughly determined as recent studies seem to indicate an oncogenic role in this pathology (Anusewicz, 2020). This needs to be further discussed as well as the--contrasting--Notch tumor suppressive role in lung small cell lung cancer (SCLC), the other main pathology in this organ.

With regard to Notch regulation by glycosylation (line 171), it should be further specified that POGLUT1 catalyzes both O-glucosylation of Notch receptors (main function) as well as O-xylosylation.

Paragraph starting in line 183: a discussion of how the pulling force may lead to exposure of the gamma-secretase site, therefore promoting this proteolytic cleavage, is necessary to further describe in more detail the Notch molecular activation process.

Gastrointestinal side effects by GSIs is discussed with respect to targets other than Notch, however, undesired effects due to on-target (Notch) inhibition in this tissue should also be considered and discussed.

In terms of synthetic lethality, it should be discuss the observation that inhibition of Notch and DDR1 with GSI LY411575 and 7rh are additive on induction of apoptosis and tumor growth blockage (Ambrogio, 2016).

Notch inhibition by fucose analogs (Schneider, 2018) needs to also de discussed as well as the potential for other analogs of other sugar residues in Notch receptors.

A more elaborated description of Ataluren mode of action in re-expressing genes carrying nonsense mutations is needed.

The work of Kobia (2014) with V‐ATPase inhibitor BafilomycinA1 (BafA1) and Pinazza (2018) in Notch inhibition needs to also be discussed.

Sentence starting in line 57 needs a better description.

Sentence starting in line 297 needs an improved description.

Sentence starting in line 368 needs a better description.

Line 423: Define CAFs

Syntax error in line 461. Update and improve the provided description.

Syntax error in line 564. Update and improve the provided description.

Typo (,) in line 619

Line 103: Edit to improve the provided description.

Line 144: A parenthesis is present and is not clear why.

Line 244: Update NCT acronym with the correct one.  Check also the entire manuscript for this abbreviation.

Line 270: typo; it should says ligands.

Line 279: TNC acronym should be corrected.

Line 293: typo (transported) needs to be corrected.

Line 305: the word secretion may lead to misunderstanding.  Update with other word(s) describing the intracellular trafficking maturation process between ER and Golgi Apparatus.

English is good.

Author Response

Dear Reviewer,

We greatly appreciate your comments and suggestions on the manuscript. These comments are very helpful to improve our work. The manuscript has been modified in line with the Reviewers’ suggestions. We hope that the revised version of the manuscript will reach the Reviewers' expectations and make the manuscript appropriate to be published in Cancers.

The changes introduced in manuscript and the list of responses to the Reviewer’s comments are listed below. The proposed changes are marked in the main text in review mode.

Open Review 2

Comments and Suggestions for Authors

Question: While Notch acts as tumor suppressor in squamous cell carcinoma in head and neck cancer (HNSCC) as well as in CSCC (cutaneous squamous cell carcinoma), the role of Notch signaling in squamous cell carcinoma of the lung remains to be more thoroughly determined as recent studies seem to indicate an oncogenic role in this pathology (Anusewicz, 2020). This needs to be further discussed as well as the--contrasting--Notch tumor suppressive role in lung small cell lung cancer (SCLC), the other main pathology in this organ.

Answer: The role of Notch signaling in various types of lung cancer was previously described by us in the articles "Shooting at Moving and Hidden Targets—Tumour Cell Plasticity and the Notch Signalling Pathway in Head and Neck Squamous Cell Carcinomas" (Kałafut et al. Cancers (Basel). 2021 Dec 10;13(24):6219. doi: 10.3390/cancers13246219) and “Context Matters: NOTCH Signatures and Pathway in Cancer Progression and Metastasis” (Misiorek et al. Cells. 2021 Jan 7;10(1):94. doi: 10.3390/cells10010094). Both of these articles were cited in the main text (Line: L224). Therefore, we have decided not to replicate this information in article.

Question: With regard to Notch regulation by glycosylation (line 171), it should be further specified that POGLUT1 catalyzes both O-glucosylation of (main Notch receptors function) as well as O-xylosylation.

Answer: Information about O-xylosylation has been added. (Based on Li et al, 2017; Structural basis of Notch O-glucosylation and O–xylosylation by mammalian protein–O-glucosyltransferase 1 (POGLUT1).

Question: Paragraph starting in line 183: a discussion of how the pulling force may lead to exposure of the gamma-secretase site, therefore promoting this proteolytic cleavage, is necessary to further describe in more detail the Notch molecular activation process.

Answer: A paragraph on the effect of pulling force and its relation to NOTCH exposure to ADAM and γ-secretase has been added to the manuscript (Based on Gordon et al, 2015; Mechanical Allostery: Evidence for a Force Requirement in the Proteolytic Activation of Notch):

“In the absence of ligand, NOTCH receptors are unable to further activation, because the activating cleavage site (S2), is in autoinhibited conformation state inside NRR. After receptor-ligand binding, DSL-NECD complex undergoes endocytosis in to signaling sending cell. This step exerting a pulling force with allows to unfolds and exposure the S2 site to proteolytic cleavage by ADAM protease (S2 cleavage; mainly ADAM10 and 17), followed by a “final cut” mediated by the γ secretase complex (γS; S3 cleavage).” Lines: L71-77

Question: Gastrointestinal side effects by GSIs is discussed with respect to targets other than Notch, however, undesired effects due to on-target (Notch) inhibition in this tissue should also be considered and discussed.

Answer: The relevant paragraph about on-target (Notch) inhibition by GSI in gastrointestinal  tract has been added to the text (based on Collins et al, 2021; Notch inhibitors induce diarrhea, hypercrinia and secretory cell Metaplasia in the human colon, and Sancho et al, 2015; Stem cell and progenitor fate in the mammalian intestine: Notch and lateral inhibition in homeostasis and disease)

„As a result, direct on-target inhibition of Notch signaling by GSI can result in severe gastrointestinal side effects, which can be persevered by additional, off-target effects. Notch signaling regulates the differentiation of progenitor cells and/or the maintenance of stem cell status in human intestinal epithelium. Consequently, blocking Notch signaling by GSI can induce clinical, endoscopic and histological abnormalities in the gastrointestinal tract, including secretory cell metaplasia of the intestinal epithelium, which in turn leads to diarrhea. Additionally, inhibition of Notch signaling by GSI results in the loss of crypt base columnar (CBC) stem cells and an increase in secretory goblet cells population at the expense of proliferating cells. Consequently, γ-Secretase can result in altered homeostasis in the gastrointestinal system by exposure to GSI, leading to rapid weight loss and even death consistent with a failure of tissue replenishment and lack of nutrient absorption.” Lines: L415-426

Question: In terms of synthetic lethality, it should be discuss the observation that inhibition of Notch and DDR1 with GSI LY411575 and 7rh are additive on induction of apoptosis and tumor growth blockage (Ambrogio, 2016).

Answer: The suggested article was included in the manuscript. The relevant paragraph has been added to the text:

„Additionally, concomitant inhibition of Notch and tyrosine kinase receptor DDR1 (Discoidin Domain Receptor 1) signaling by LY411575 and 7rh, respectively, was demonstrated to trigger blockade of cell growth and to induce apoptosis in KrasG12V-driven tumors via modulation of MAPK signaling. Additionally, the combination of LY411575 and 7rh induced higher apoptosis than either agent alone. This shows the potential of combining Notch inhibitors with other compounds to achieve synthetic lethality phenomenon.” Lines: L484-490

Question: Notch inhibition by fucose analogs (Schneider, 2018) needs to also de discussed as well as the potential for other analogs of other sugar residues in Notch receptors.

Answer: The suggested article was included in the manuscript. The relevant paragraph has been added to the text:

„Notch signaling can also be modulated by modifying the sugar residues of EGF-like extracellular domains of the NECD. Inhibition of Notch signaling can be accomplished with O-fucose analogs such as 6-alkynyl and 6-alkenyl fucose, specifically, by incorporating these analog into EGF-like domain 8 (EGF8), which disrupts the DLL-NECD interaction. Interestingly, both of analogs not disturb Jagged-induced, Notch signaling. This observation opens up a new field of study for different O-glycans to fine-tune the strength of Notch signaling by modification sugar residues of the NICD.” Lines: L373-379

Question: A more elaborated description of Ataluren mode of action in re-expressing genes carrying nonsense mutations is needed.

Answer: More specific Ataluren mode of action was described in text (based on Welch et al, 2007; PTC124 targets genetic disorders caused by nonsense mutations; Siddiqui et al, 2016; Proposing a mechanism of action for ataluren; Tutone et al, 2019; Deciphering the Nonsense Readthrough Mechanism of Action of Ataluren: An in Silico Compared Study; and Huang et al, 2022; Ataluren binds to multiple protein synthesis apparatus sites and competitively inhibits release factor-dependent termination):

„PTC-124  selectively induces ribosomal readthrough of premature but not normal termination (stop) codons. Mechanistically, PTC-124 acts by binding to rRNA, which inhibits of ribosomal activity to mRNA termination. Consequently, PTC-124 allows to "bypass" mostly UGA nonsense mutations in the mRNA. This makes it possible to synthesis of a full-length protein with a changed amino acid residue at the mutation site.” Lines: L699-704

Question: The work of Kobia (2014) with V‐ATPase inhibitor BafilomycinA1 (BafA1) and Pinazza (2018) in Notch inhibition needs to also be discussed.

Answer: The above-mentioned publications were added to the manuscript:

„Similarly, Bafilomycin A1 (BafA1), a highly specific Vacuolar H+ ATPase (V‐ATPase) inhibitor, leads to accumulation of NOTCH in the endo‐lysosomal system, and consequently, reduces Notch signaling in normal breast cells. Interestingly, in  breast cancer model, BafA1 treatment reduces the growth in cells expressing membrane tethered, constitutively active NOTCH1 forms, while sparing cells expressing cytoplasmic forms.” Lines: L344-349

Additionally, Pinazza et al. (DOI https://doi.org/10.1038/s41388-018-0234-z) has also been cited for Trichostatin A activity in case of T-ALL.

Questions (grammar errors and typos):

Sentence starting in line 57 needs a better description.

Sentence starting in line 297 needs an improved description.

Sentence starting in line 368 needs a better description.

Answer: The sentences have been corrected. The proposed changes are marked in the main text in review mode:

Questions (grammar errors and typos):

Line 423: Define CAFs

Syntax error in line 461. Update and improve the provided description.

Syntax error in line 564. Update and improve the provided description.

Typo (,) in line 619

Line 103: Edit to improve the provided description.

Line 144: A parenthesis is present and is not clear why.

Line 244: Update NCT acronym with the correct one.  Check also the entire manuscript for this abbreviation.

Line 270: typo; it should says ligands.

Line 279: TNC acronym should be corrected.

Line 293: typo (transported) needs to be corrected.

Line 305: the word secretion may lead to misunderstanding.  Update with other word(s) describing the intracellular trafficking maturation process between ER and Golgi Apparatus.

Aswer: The text has been corrected and redacted as suggested by Reviewer 2. The sentences and acronyms have been corrected and improved, errors and typos removed and the manuscript reviewed again.

Comments on the Quality of English Language

English is good.
